# Fused Filament Fabricated Polypropylene Composite Reinforced by Aligned Glass Fibers

**DOI:** 10.3390/ma13163442

**Published:** 2020-08-05

**Authors:** Eugene Shulga, Radmir Karamov, Ivan S. Sergeichev, Stepan D. Konev, Liliya I. Shurygina, Iskander S. Akhatov, Sergey D. Shandakov, Albert G. Nasibulin

**Affiliations:** 1Laboratory of Nanomaterials, Center for Photonics and Quantum Materials, Skolkovo Institute of Science and Technology, Moscow 121205, Russia; 2Center for Design, Manufacturing & Materials, Skolkovo Institute of Science and Technology, Moscow 121205, Russia; Radmir.Karamov@skoltech.ru (R.K.); I.Sergeichev@skoltech.ru (I.S.S.); s.konev@skoltech.ru (S.D.K.); I.Akhatov@skoltech.ru (I.S.A.); 3Department of General and Experimental physics, Kemerovo State University, Kemerovo 650043, Russia; 437chem@kemsu.ru (L.I.S.); sergey.shandakov@gmail.com (S.D.S.); 4Department of Chemistry and Materials science, Aalto University, P.O. Box 16100, FI-00076 Espoo, Finland

**Keywords:** 3D printing, short glass fibers, FFF, polypropylene, fiber orientation, micro CT

## Abstract

3D printing using fused composite filament fabrication technique (FFF) allows prototyping and manufacturing of durable, lightweight, and customizable parts on demand. Such composites demonstrate significantly improved printability, due to the reduction of shrinkage and warping, alongside the enhancement of strength and rigidity. In this work, we use polypropylene filament reinforced by short glass fibers to demonstrate the effect of fiber orientation on mechanical tensile properties of the 3D printed specimens. The influence of the printed layer thickness and raster angle on final fiber orientations was investigated using X-ray micro-computed tomography. The best ultimate tensile strength of 57.4 MPa and elasticity modulus of 5.5 GPa were obtained with a 90° raster angle, versus 30.4 MPa and 2.5 GPa for samples with a criss-cross 45°, 135° raster angle, with the thinnest printed layer thickness of 0.1 mm.

## 1. Introduction

The fused filament fabrication (FFF) technique is one of the additive manufacturing processes where filaments are fused together to fabricate a solid part [1]. Despite the technique originally being used for rapid prototyping and manufacturing of decorative parts, nowadays, with advancements in printers and raw materials, the FFF technique has been widely adopted for the manufacturing of rapidly customizable structural parts [2,3]. For such adoption, additively manufactured parts should be manufactured with an appropriate strength level. However, the pure thermoplastics most used in FFF lack the strength of produced functional and load-bearing parts. These drawbacks restrict wide applications of the FFF technique [4,5].

The strength and rigidity of polymer materials can be significantly reinforced by introducing fibers. For additive manufacturing applications, either continuous [6,7] or short chopped carbon and glass fibers [8] can be utilized. Continuous fibers require printers with special construction and software, and their usage is not commonplace, while short fibers are widely used to reinforce polymers and to improve printability by minimizing the shrinkage along printed layers [9].

The mechanical properties of short fiber-reinforced composites, first of all, greatly depend on fiber orientation and length distributions [10,11,12]. The concentration and orientation of short carbon fibers in FFF based on acrylonitrile butadiene styrene (ABS) were investigated by Tekinalp et al. [13]. It was found that the mean fiber length decreased with the loading amount during compounding, due to the damage caused by high shear stress. Therefore, at 10% loading the mean fiber length was around 350 micrometers and only around 200 micrometers at 40%. These correspondingly resulted in 50 and 60 MPa ultimate tensile strengths of the printed test parts. They also concluded that during the FFF process the fibers are more oriented and uniformly dispersed in the printed part, when compared to a compression molding method. However, this did not result in higher tensile strength for the printed specimens, which was attributed to the presence of voids in the printed parts. Therefore, it is critical to characterize the inner structure of the printed specimens, including volume fraction and morphology of voids, along with fiber orientation.

The influence of the raster angle (orientation of the printing lines) was demonstrated by Ziemian et al. [14], who found that the highest ultimate tensile strength (UTS) and elasticity modulus were achieved for a 90° raster angle (when all printed lines are parallel and match the pulling force direction during the tensile test). This could be explained as each extruded line being treated as a separate fiber, which can take higher stress in the fiber direction and the lowest stress across the fiber direction [15].

Printed layer thickness most likely affects the number of voids in the printed structure. This could be concluded from the fact that thinner layers usually result in a denser structure with fewer voids. In work [16], the tensile strength of the FFF printed parts was increased by decreasing the layer thickness (in the range of 0.10 to 0.30 mm, which corresponds to 25% and 75% of the printer nozzle diameter).

Polypropylene (PP) is one of the promising materials for use in the FFF process because it is a widely available, cheap, and nontoxic polymer with good recyclability. However, its crystalline nature and tendency toward warping make PP not always suitable or popular in the 3D printing community. A few experimental papers recently examined the limitations of printing with unfilled PP [17,18]. A potential solution to prevent printed parts from shrinking and warping is to incorporate fillers, such as short glass fibers [19,20]. However, little attention has been devoted to the influence of fiber orientation on the mechanical properties of the printed parts.

In our work, we investigated the influence of printed raster angle and layer thickness on reinforcing fiber alignment and the resulting tensile strength and modulus of the printed composite parts. For the first time the fiber’s alignment degree was calculated and visualized with the help of X-ray micro-computed tomography (micro-CT). The results show good fiber alignment within the printed line and a great impact of fiber alignment on the resulting tensile strength of the printed parts.

## 2. Materials and Methods

### 2.1. Test Material and Specimen Fabrication

In this work we used polypropylene filament reinforced by short glass fibers. Table 1 summarizes the physical properties of the filament.

Samples for mechanical testing were fabricated using a conventional desktop 3D printer (Sapphire S, TWO TREES, Shenzhen, China), modified to have a closed chamber to maintain a steady temperature of 35 °C.

The specimens were built for tensile tests according to ISO 527-4, with a constant thickness of 4 mm (Figure 1a) with two types of infill raster angles: the printed layers are printed alternately with 45° and 135° sequences, henceforth referred to as (45, 135) (Figure 1b), and 90° and 90° sequences, where all printing lines are parallel, henceforth referred to as (90, 90) (Figure 1c). Printed specimens had different numbers of layers: specimens with a thickness of 0.1 mm had 40 layers, 0.15 mm had 26 layers, 0.2 mm had 20 layers, 0.25 mm had 16 layers, and 0.30 mm had 13 layers. All samples were measured before tensile testing to get correct values of the cross-section area.

We did not prepare specimens with a 180° raster orientation because this would provide the weakest tensile strength results, due to the great influence of the fusion between adjacent layers [16]. The printing layer thickness was varied from 0.10 to 0.30 mm. The following settings were used: “wall line count” of 2, “top” and “bottom thickness” of 0, “infill pattern”—“lines”, and “print cooling disabled”. The specimens were sliced using an Ultimaker “Cura” slicer (Ultimaker B.V., Utrecht, The Netherlands).

All test specimens were fabricated with the following printing parameters: a printer nozzle diameter of 0.6 mm, a printhead temperature of 260 °C, a heatbed temperature of 95 °C, a printing speed of 50 mm/s, and infill density of 100%. Temperature inside the printer chamber was 35 °C.

### 2.2. X-ray Micro-Computed Tomography

X-ray micro-computed tomography (micro-CT) image processing gave a high-fidelity view of phenomena which are normally difficult to calculate, such as fiber length, fiber orientation distributions, voids, etc. [21,22]. In the present study, the micro-CT technique was utilized to analyze the microstructure of the printed material and calculate fiber orientation distributions.

For the micro-CT characterization, cylinders of 4 mm diameters were cut from the tensile specimens by CNC machine and X-ray scanned in high resolution (2.2 μm/pix).

The structure of the specimens was analyzed by a GE Phoenix CT System v|tome| × L240 (Waygate Technologies, Cincinnati, OH, USA). The system is equipped with long life open micro-300 kV and nanofocus 180 kV X-ray tubes and a 2048 × 2048 pixel^2^ on 14-bit GE DXR-250 flat panel detector. Nanofocus X-ray tubes were used, the beam current and the accelerating voltage of the X-ray tube were 220 µA and 60 kV, respectively. The number of projections was 2400 and the detector integration time was 0.8 s. A molybdenum target, suitable for weak absorbing specimens, was used.

The original volume size of the obtained micro-CT images was 1620 × 1620 × 1620 pixel^3^ (3524 × 3524 × 3524 μm^3^). The volume was analyzed with VoxTex software (Vortex Studio 2019b, CM Labs Simulations, Montreal, QC, Canada) [23], based on a structure tensor analysis method. The method allows calculation of fiber orientation distributions with sufficient accuracy even from low resolution scans [24].

### 2.3. Mechanical Testing

The tensile tests were performed according to the ISO 527-4 standard by an Instron 5969 testing machine (Instron, Norwood, MA, USA) with a loading speed of 1 mm/min and a load cell capacity of 50 kN. All samples were measured before tensile testing to get correct values of the cross-section area. The strains were measured utilizing a digital image correlation (DIC) system from Correlated Solutions INC, Irmo, SC, USA. The pictures were captured every 1 s or load increment 3 N, whichever happened first. The load at break was recorded by the load cell from a point of a rapid load drop. The strain at break was extracted from the last DIC picture where the specimen was still one part. Six specimens were tested per group and, therefore, the total number of tested specimens was 60. The test groups were identified by their print pattern and layer thickness.

Six specimens were tested per sample with a specific printed pattern and layer thickness. The medians of the tensile stress-strain curves were plotted for every sample. The scatter given in Table 2.

## 3. Results and Discussion

### 3.1. Fibers Orientation

Figure 2 shows a 3D visualization model made with the help of micro-CT technique: one cubic millimeter of a sample printed with a 0.10 mm layer thickness and a raster angle of (45, 135), where 45° layers alternate with 135° layers. Reinforcing short glass fibers with an average length of 250 micrometers and a high degree of orientation in the direction of the nozzle movement can be clearly identified.

To understand how fiber orientation depends on printing parameters, we prepared samples with two types of raster angles, of (45, 135) and (90, 90), and with various printed layer thicknesses, from 0.10 to 0.30 mm. Fiber orientation distributions were calculated in a polar coordinate system, where the orientation of the fibers can be described by two angles: in-plane φ-angle and out-of-plane θ-angle (Figure 3a).

The in-plane angle—φXY (Figure 3b,c,e) corresponds to the printing direction of the printhead nozzle and mainly lies in the range of ±10° for both the raster configurations of (45, 135) and (90, 90) for a given range of printed layer thicknesses from 0.10 to 0.30 mm. At the same time, the out-of-plane fiber orientation distribution (θ-angle) is not as emphasized for the thicker 0.30 mm layers, when compared to the thinnest 0.10 mm layers, for both raster configurations (Figure 3d).

Therefore, we can conclude that the fibers are mostly aligned with the nozzle movement direction, regardless of printing layer thickness. This happens when fibers are extruded along with the polymer matrix and are confined between the previous printed layer and the printing nozzle. For the thinner layers a notably higher degree of fiber orientation and more uniform distribution in the out-of-plane (θ-angle) direction can be observed. This is confirmed by the micro-CT analysis (Figure 4a,b).

### 3.2. Mechanical Properties

Figure 5a shows the stress–strain curves of the specimens printed with varying layer thicknesses for the (90, 90) raster. The ultimate tensile strength of the printed materials revealed good values for the 3D printed polypropylene composite, which varied from 44 to 57 MPa, where the highest value corresponds to the thinnest layer thickness of 0.10 mm. This is roughly 48% higher than the best value obtained with the (45, 135) raster. Such a high maximum ultimate tensile strength value most probably comes from the relatively long reinforcing glass fibers, which are around 0.25 mm long. The fiber alignment along the direction of the applied force is an additional factor which helps effectively transfer the load and makes a significant contribution to the ultimate tensile strength of the (90, 90) raster samples [25].

It is interesting to note that for the (90, 90) raster samples, ultimate tensile strength drops with increasing layer thickness (Figure 5b) and practically does not change starting with a layer thickness of 0.20 mm. We attribute this behavior to the mean fiber length, which is around 0.25 mm and longer than the layer thickness. Particularly, fibers longer than the printing layer showed better alignment during printing.

Figure 5c shows the stress–strain curves of the specimens printed with varying layer thicknesses for the (45, 135) raster. Samples with this type of raster angle demonstrated significantly higher strain to failure values (with the yield point) when compared with the samples with the (90, 90) raster. We attribute this phenomenon to the much less pronounced orientation of short glass fibers in the in-plane directions (which leads to samples with less tensile modulus), to larger strain at break values.

It is worth mentioning that ultimate tensile strength and modulus values of the (45, 135) raster samples do not exhibit any dependence on printing layer thicknesses and are around 29 and 2200 MPa, respectively (Figure 5d and Table 2). As was previously mentioned, the more the reinforcing fiber direction coincides with the tensile load vector, the larger the tensile strength. In the case of the (45, 135) raster, significantly larger fiber misalignment in the in-plane direction (compared to the (90, 90) raster) negates the effect of fiber alignment in the out-of-plane direction and diminishes the effect, which comes from the variation in the printing layer thickness. This leads to a 53% lower tensile modulus best value for the (45, 135) raster, compared to the (90, 90) raster.

The strain at break values for both types of raster angles are represented in Table 2. For the (45, 135) raster it lies between 0.07 and 0.05 mm/mm for 0.10 and 0.30 mm layer thickness, respectively. For the (90, 90) raster it is almost two to three times smaller and lies between 0.025 and 0.015 mm/mm for 0.10 and 0.30 mm layer thicknesses, respectively.

Much higher strain at break values for the (45, 135) raster could be attributed to the previously-discussed orientation of the short glass fibers’ in-plane and out-of-plane directions, according to Cordin et al. [26] and the so-called modified rule of mixture (this rule suggests a strong orientation dependence of composite stiffness). Particularly, the more the angle of the fibers deviates from 0° and approaches a ±  45° angle, the softer the composite becomes, making the materials easier to deform and thus increasing the strain-at-break value.

## 4. Conclusions

This work presents the results of micro-CT analysis and mechanical testing of specimens produced by 3D printing with the FFF method, using polypropylene filament reinforced by short glass fibers. The specimens were printed with raster angle orientations of (90, 90) and (45, 135), with varying printing layer thicknesses of 0.10 to 0.30 mm. For the in-plane projection, all printed samples demonstrate good fiber orientation along the moving direction. The best alignment in the out-of-plane projection was represented by the samples with the smallest layer thickness used (0.10 mm).

For the ultimate tensile strength of both types of raster angles, the best results were obtained from a layer thickness of 0.10 mm, with a mean value of 57.4 MPa. This was around 48% higher for the parallel raster orientation of (90, 90), when compared to the angle of (45, 135). The tensile modulus for the raster angle of (90, 90) was also 53% higher than that of the angle of (45, 135) for the 0.10 mm layer thickness.

For the samples printed with the raster direction of (90, 90), the influence of the layer thickness gets less pronounced with thicknesses above 0.20 mm. We believe this can be closely correlated to the mean fiber length (which, in our case, was around 250 microns). The results show the significant impact of short glass fiber orientation on the tensile strength, modulus, and stress at break of the printed specimens. For practical applications, this means that better tensile strength of FFF manufactured short fiber-reinforced parts could be obtained along the printing direction, with a layer thickness smaller than the mean fiber length.

## Figures and Tables

**Figure 1 materials-13-03442-f001:**
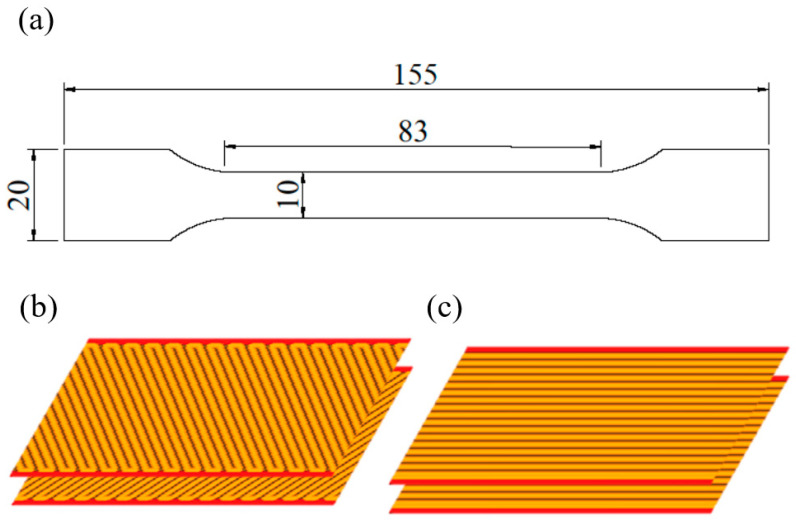
(**a**) The geometry of the tensile test specimen with printing raster angles of (**b**) (45, 135) and (**c**) (90, 90).

**Figure 2 materials-13-03442-f002:**
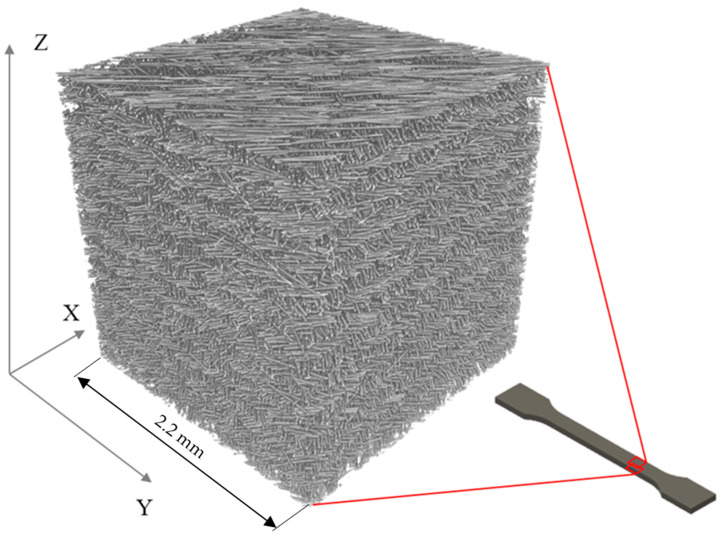
3D visualization of the micro-CT image of a specimen with a raster angle of (45, 135) and a size of roughly 10 mm^3^ (10,003 pixel^3^ volume), printed with a 0.10 mm layer thickness.

**Figure 3 materials-13-03442-f003:**
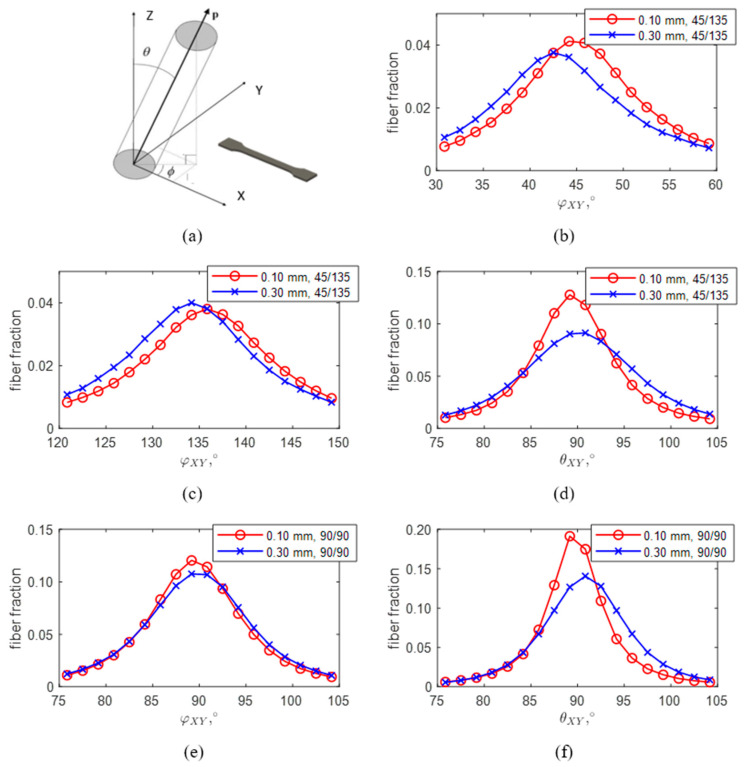
(**a**) Polar coordinate system, where the orientation of fibers are described by two angles: in-plane φ-angle and out-of-plane θ-angle. Fiber orientation distributions obtained for the specimens with a raster angle of (45, 135) and thicknesses of 0.10 mm (red) and 0.30 mm (blue), (**b**) 45° in-plane orientation, (**c**) 135° in-plane orientation, and (**d**) out-of-plane orientation. Fiber orientation distributions obtained for the specimens with a raster angle of (90, 90) and thicknesses of 0.10 mm (red) and 0.30 mm (blue), (**e**) in-plane orientation and (**f**) out-of-plane orientation.

**Figure 4 materials-13-03442-f004:**
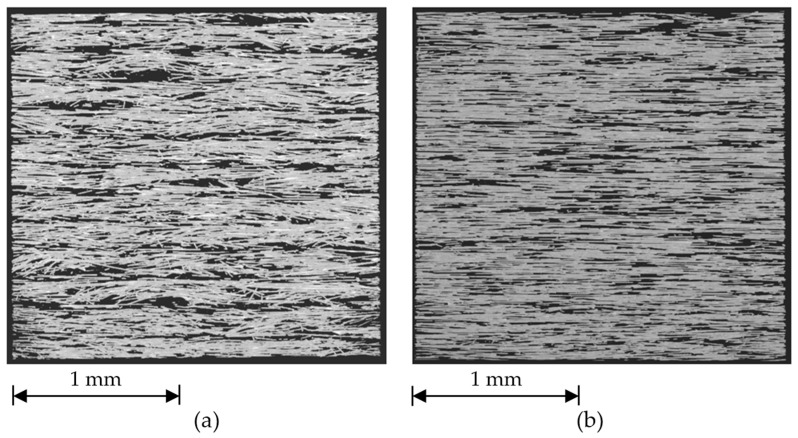
Three-dimensional visualization of longitudinal sections (depth of 150 µm) in the XZ plane of (90, 90) specimens printed with layer thicknesses of (**a**) 0.30 mm and (**b**) 0.10 mm.

**Figure 5 materials-13-03442-f005:**
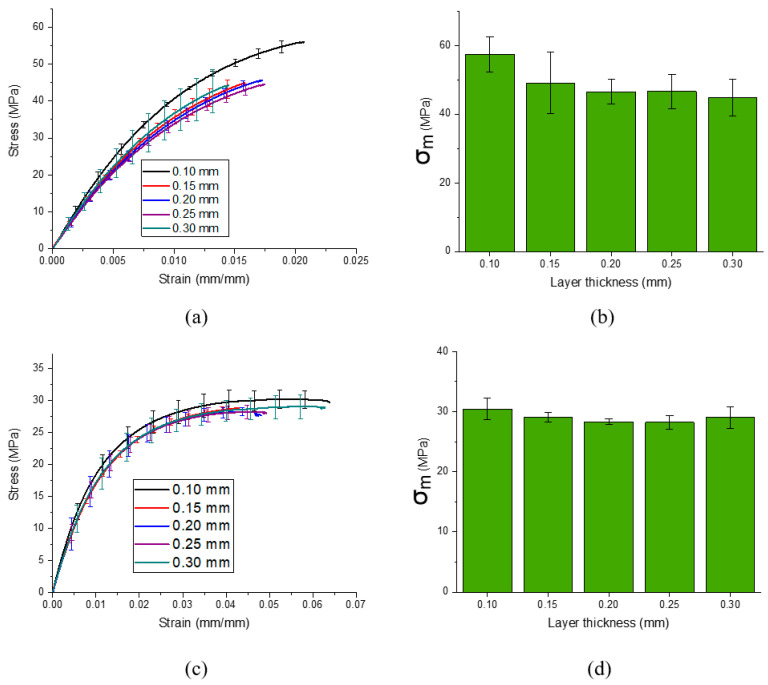
Results of mechanical tests: (**a**,**c**) stress–strain curves of the printed samples with raster angles of (90, 90) and (45, 135), respectively; (**b**,**d**) bar charts representing ultimate tensile strength, dependent on layer thickness, for the samples with raster angles of (90, 90) and (45, 135), respectively.

**Table 1 materials-13-03442-t001:** Physical properties of polypropylene filament reinforced by short glass fibers.

Properties	Values	Unit
Specific gravity	1.07	g/cm^3^
Melting temperature	145	°C
Glass fiber length	100–300	µm
Glass fiber diameter	15–20	µm
Glass fiber average mass fraction	25	%
Diameter tolerance	±0.05	mm

**Table 2 materials-13-03442-t002:** Results of the mechanical tests of the 3D printed samples.

Layer Thickness (mm)	Raster Angle	Modulus (MPa)	Strength (MPa)	Stress at Break (MPa)	Strain at Break (mm/mm)
0.10	(45, 135)	2470 ± 127	30.4 ± 1.8	26.8 ± 4.7	0.073 ± 0.021
0.15		2080 ± 125	29.1 ± 0.8	26.6 ± 1.2	0.067 ± 0.031
0.20		2250 ± 668	28.3 ± 0.5	26.0 ± 4.6	0.055 ± 0.022
0.25		2230 ± 298	28.2 ± 1.1	25.1 ± 2.4	0.053 ± 0.037
0.30		2190 ± 453	29.0 ± 1.8	25.7 ± 3.9	0.074 ± 0.072
0.10	(90, 90)	5280 ± 345	57.4 ± 5.1	54.5 ± 8.1	0.025 ± 0.011
0.15		4610 ± 291	49.1 ± 8.9	46.2 ± 7.6	0.022 ± 0.011
0.20		4510 ± 415	46.5 ± 3.6	43.1 ± 1.6	0.018 ± 0.003
0.25		4440 ± 198	46.6 ± 5.0	42.6 ± 7.0	0.021 ± 0.007
0.30		4830 ± 708	44.8 ± 5.4	42.1 ± 8.9	0.015 ± 0.002

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
