# Peer review of "Fused Filament Fabricated Polypropylene Composite Reinforced by Aligned Glass Fibers"

_materials, 2020, doi:10.3390/ma13163442_

Round 1
Reviewer 1 Report
Review report
Manuscript ID: materials-861667
Title: Fused Filament Fabricated Polypropylene Composite Reinforced by Aligned Glass Fibers
In this article, the authors have studied the effect of fibre orientation on the mechanical properties of glass fibre reinforced PP composite, which is fabricated via fused filament fabrication technique. The influence of printed layer thickness and raster angle on the fibre orientation, was discussed in the manuscript. Overall, the manuscript is well written and results are well explained. However there are some minor points which need to be clarified:
- Discussion on relevant literatures on PP composites fabricated via fused filament fabrication method is missing in the introduction part.
- How did you fabricate PP/glass fibre composites? It would be good to mention about the fabrication process of composites in detail, i.e. how much fibre loading and what conditions have been used for the fabrication.
Reviewer 2 Report
The manuscript “Fused Filament Fabricated Polypropylene Composite Reinforced by Aligned Glass Fibers” reports the tensile properties of PP-glass fiber composites fabricated via 3D printing method. The novelty of work is not clear and the results and discussions are not sufficient besides the manuscript was not written well. Therefore, the reviewer can not recommend it for the further review process.
Comments:
- Line 8: “2 Aalto University, P.O” must be changed into “3 Aalto University, P.O”
- The abstract should state briefly the purpose of the research, the principal results, and major conclusions Therefore the abstract should be rewritten and lines 11to 14 can be removed.
- Line 28: parts[2], [3] must be changed into [2,3]. And some others.
- The last paragraph of the Introduction section must be edited to clearly shows the novelty of work and the procedures. It may be supported by a schematic figure.
- Section 2. All the sources of the materials must be introduced.
- Section 2, it should be clearly identified how the filaments were prepared.
- The specification of the tensile machine must be presented in the manuscript along with the test condition i.e. crosshead speed, the capacity of the load cell, and …
- The prepared samples must be also evaluated by Dynamic Mechanical Thermal Analysis (DMTA) at the tensile mode.
- The SEM figures of the cross-section of the fractured surfaces must be provided.
- Statistical analysis must be carried out to validate the significant differences between the results.
- Discussions are totally missed in the results and discussions section. The authors just reported the obtained results. The results must be compared with similar previous published results and the reasons for differences must be explained.
- The information on the serval references is missed! See references 3, 4, and ….
Reviewer 3 Report
This paper is aiming to study the impact on the tensile strength and modulus of the printed dogbones using glass fiber reinforced PP by the raster angle and layer thickness. The goal of the study in my opinion is too narrow and too specific to benefit the 3D printing community. Below are a few detailed issues regarding this manuscript.
- It might be better to have a graph to explain or define the “raster angle” in addition to the line of 5-words
- In the section of Materials and Methods:
- Can the authors explain why polypropylene was chosen? PP does not seem to be the most commonly used 3D printed materials. The authors also need to provide the supplier of this material.
- What’s the printing profile like for the tensile specimens? Were there any “walls” in the printed specimens? These could impact the tensile results as well.
- How many layers in the printed tensile specimens? What’s the thickness of the specimens? Was the number of the layers kept constant across all the samples or was the total thickness of the samples kept constant?
- There was no descriptions regarding how the tensile test was done. Specifically, how many samples? What was the tension rate? How was the standard deviations calculated? Why there were standard deviations on the stress-strain curves? Were the curves shown in Fig. 5a results of “average” of several tensile curves? Can the authors explain how “average” tensile curves were produced?
- What’s the y-axis in Figure 3b-f? It was not intuitive enough for the audience to understand.
- Please add scale bars to Figure 2&4
- Regarding Line 160 to 161: how did the authors get this conclusion? Please add the citations/references or more explanations.
- Figure 5: can the authors explain:
- Why there was a yield point in Fig 5c?
- Why [45,135] has higher strain than [90,90]?
- Figure 5b the bar value (~50MPa) for 0.15mm layer does not seem to match the result from Figure 5a for the same layer (~45MPa). Can the authors double check?
- Also, in my opinion, samples printed in the 180Ëš raster angles should not be ignored. After all, the quality of a print from FFF printer depends on the interlayer adhesion, which can only be directly studied from the samples printed in the 180Ëš raster angles. Without the study of those samples, the results of the manuscript is not meaningful enough to benefit the 3D printing community.
If the authors can address all above issues, the manuscript might be able to be considered for publication.
Round 2
Reviewer 2 Report
The manuscript can be accepted.
Author Response
Thank You for the review! We appreciate Your help in making the results of our research available to the community.
Reviewer 3 Report
The revised version has addressed some of the issued stated in the previous review but unfortunately there are still a few critical outstanding ones that disqualify this manuscript to be ready for publication.
- The design of the dogbone samples: Though the authors have provided the thickness of the layers, they did not provide the how many layers per sample for different layer thickness. What is held constant for the dogbone samples printed with different layer thickness? Was the total thickness of the dogbone samples held constant? If so then samples with different printed layer thickness would have different numbers of layers? If so then the authors would need to also consider how the numbers of layers impact the tensile results.
- The tensile results: The authors did not make it clear that how did they "average" the tensile data. Usually tensile curves can not be averaged but being reported using the median of the curves. Please make it clear to the audience; or provide all the raw stress-strain curves in the supplemental docs. In addition, the authors did not answer the question that how many specimen per sample (with a specific printed layer thickness) were tested.
- I think it is include the authors' answer to 6.1 (Why there was a yield point in Fig 5c?) and 6.2 (Why [45,135] has higher strain than [90,90]?) in the manuscript.
If the above three can be addressed this paper could be reconsidered for publication.
Round 3
Reviewer 3 Report
Only one comment on the most recent revised manuscript, the factor of number of layers in the printed dogbones should be also studied (or at least needs to be mentioned in the manuscript). Other than that, everything else is much improved.
